# Licuri Kernel (*Syagrus coronata (Martius) Beccari*): A Promising Matrix for the Development of Fermented Plant-Based Kefir Beverages

**DOI:** 10.3390/foods13132056

**Published:** 2024-06-27

**Authors:** Janaína de Carvalho Alves, Carolina Oliveira de Souza, Livia de Matos Santos, Suelen Neris Almeida Viana, Denilson de Jesus Assis, Pedro Paulo Lordelo Guimarães Tavares, Elis dos Reis Requião, Jéssica Maria Rio Branco dos Santos Ferro, Mariana Nougalli Roselino

**Affiliations:** 1Northeast Biotechnology Network, Institute of Health Sciences, Federal University of Bahia, Av. Reitor Miguel Calmon, s/n, Salvador 40231-300, Brazil; 2Graduate Program in Food Science, Faculty of Pharmacy, Federal University of Bahia, R. Barão de Jeremoabo, 147, Salvador 40170-115, Brazil; liviamatos@ufba.br (L.d.M.S.); suelen.neris@ufba.br (S.N.A.V.); pp.lordelo@gmail.com (P.P.L.G.T.); 3College of Pharmacy, Federal University of Bahia, R. Barão de Jeremoabo, 147, Salvador 40170-115, Brazil; elis.requiao@ufba.br (E.d.R.R.); mariana.roselino@ufba.br (M.N.R.); 4School of Exact and Technological Sciences, Salvador University, Av. Tancredo Neves, 2131, Salvador 41820-021, Brazil; denilson.assis@unifacs.br; 5Graduate Program in Chemical Engineering (PPEQ), Polytechnic School, Federal University of Bahia, R. Prof. Aristídes Novis, 2, Salvador 40210-630, Brazil; 6Postgraduate Program in Microbiology (PPG-MICRO), Institute of Biology, Federal University of Bahia, R. Barão de Jeremoabo, 668, Salvador 40170-115, Brazil; jessica.ferro@ufba.br

**Keywords:** kefir, *Lactobacillus*, metagenomics, fermentation, non-dairy alternative

## Abstract

New licuri-based kefir beverages were obtained using water kefir grains as fermentation inoculum (1, 2.5, and 5%) under different fermentation times (24 and 48 h). Metagenomic sequencing of the kefir grains adapted to the aqueous licuri extract revealed *Lactobacillus hilgardii* and *Brettanomyces bruxellensis* to be predominant in this inoculum. The excellent adaptation of the kefir grains to the licuri extract raised the possibility of prebiotic action of these almonds. The beverages showed acidity values between 0.33 ± 0.00 and 0.88 ± 0.00 mg lactic acid/100 mL and pH between 3.52 ± 0.01 and 4.29 ± 0.04. The viability of lactic acid bacteria in the fermented beverages was equal to or greater than 10^8^ CFU/mL, while yeasts were between 10^4^ and 10^5^ CFU/mL. There were significant differences (*p* < 0.05) in the proximate composition of the formulations, especially in the protein (1.37 ± 0.33–2.16 ± 0.84) and carbohydrate (5.86 ± 0.19–11.51 ± 1.26) contents. In addition, all the samples showed good stability in terms of acidity, pH, and viability for LAB and yeasts during 28 days of storage (4 °C). Overall, the beverages showed a dominant yellow-green color, non-Newtonian pseudoplastic behavior, and high mean scores in the sensory evaluation. This study provided evidence of the emerging potential of licuri in the plant-based beverage industry.

## 1. Introduction

Plant-based beverages refer to products made exclusively from plant sources. This segment is booming and has generated much income in recent years. According to Allied Market Research [1] data, the plant-based beverage market’s annual growth rate will be 6.7% by 2026. 

Although this phenomenon was initially motivated by the increase in consumers who are vegan, intolerant, or allergic to animal products, nowadays, given the climatic conditions and environmental impacts caused by the production of animal milk, the search for alternative, sustainable, and eco-efficient foods has become a collective interest [2,3].

However, as an alternative to dairy products, plant-based beverages still face certain limitations, mainly due to their lower protein content [4] and anti-nutritional compounds in cereals and legumes, such as saponins, phytates, tannins, and enzyme inhibitors. These compounds compromise nutrient absorption and impart bitter or astringent tastes that negatively affect the sensory properties of these products [5,6]. 

In this sense, fermentation has been shown to be effective in optimizing the quality of plant-based beverages. Studies have reported a reduction in anti-nutritional compounds such as phytic acid and raffinose in lentil-based beverages [7], resulting in a more favorable nutritional profile. In addition, from a sensory perspective, the metabolites produced during fermentation, such as lactic acid and acetic acid, can mask unpleasant tastes and odors. This effect has been observed in soy [8], lupin, and pea beverages [9], where fermentation with different strains of the Lactobacillus genus significantly improved the sensory profile, making these beverages more palatable and acceptable to consumers.

Water kefir grains have historically been used as a starter culture in fermentation processes. They are gelatinous and irregularly shaped and consist of a dextran-type polymer matrix produced extracellularly by lactic acid bacteria using sucrose as their main substrate. In addition, yeasts and acetic acid bacteria are also found in these grains. However, the microbial profile varies depending on the geographical origin, substrate, and fermentation conditions [10,11]. 

There is no definitive archaeological evidence on the origin of water kefir [12]. The way it is obtained is still mainly through personal donation [11], since to date there has been no success in recombining the defined and isolated strains to reconstitute the grains in vitro [13].

To promote fermentation, water kefir grains are inserted into the solution to be fermented, where they are incubated at the appropriate temperature and time. At the end of the cycle, the grains are separated from the substrate, where they are available to start a new fermentation or to be stored under refrigeration [11]. 

The beverage fermented by kefir grains has a pleasant taste and aroma due to the organic acids, ethanol, carbon dioxide, and volatile compounds produced during fermentation [11,14,15,16]. In addition, it is worth noting that products fermented by kefir grains have a potential functional effect due to the presence of probiotic microorganisms that promote the balance of the intestinal microbiota and are consequently associated with significant improvements in consumer health [17].

Traditionally, water kefir grains are grown in a sucrose solution, but studies have shown that they adapt to different plant extracts, such as walnut extract [18,19], soy [20], linseed [21], hazelnut [22,23], black carrot, blackberry and strawberry [24], coconut [25], pitaya [16], and fig [26].

Considering this, *Syagrus coronata* (Martius) Beccari, popularly known as licuri, is found in the semi-arid regions of Brazil and has emerged as a new substrate for fermenting kefir grains. Since 2018, this palm tree has been recognized as an intangible heritage of Bahia [27], and the kernel of its fruit has important nutritional value, composed of 49% lipids, 11% proteins, 10% carbohydrates, and 1.2% fibers [28]. This makes it an excellent substrate for fermented beverages as it provides the nutrients to promote fermentation by water kefir grains and bioactive compounds that may have additional health benefits. Studies have shown that the licuri kernel has antibacterial, antiparasitic, antioxidant, and hypoglycemic activities [29,30,31,32]. 

In addition, licuri has a characteristic flavor that can complement and enrich the sensory profile of a fermented beverage [33,34]. Expanding the field of application of licuri is also in line with the objectives of sustainable development [2] since this fruit’s extraction is carried out by economically vulnerable Brazilian populations. Licuri is considered a Slow Food stronghold, which means there is an ongoing effort to promote the preservation, revival of traditions, promotion, dissemination, and strengthening of the marketing of products derived from licuri. 

Therefore, this study aimed to develop a fermented beverage using water kefir grains and aqueous licuri extract. Metagenomic analysis was conducted on the kefir grains adapted to the aqueous licuri extract. The impact of the fermentation time and inoculum quantity variables on the physicochemical parameters, microbiological viability, and nutritional, rheological, and sensory properties of the beverages was also verified.

## 2. Materials and Methods

### 2.1. Materials

The water kefir grains were provided by the Laboratory for Studies in Food Microbiology (LEMA) located at the Faculty of Pharmacy of the Federal University of Bahia (UFBA), Salvador-Bahia, Brazil. The licuri kernels were obtained from the Cooperativa de Produção da Região do Piemonte da Diamantina (COOPES), located in the city of Capim Grosso-Bahia, and transported to LEMA, where they were sanitized (chlorinated water at 200 ppm for 30 min) and stored at −20 °C. The demerara sugar (União^®^) was purchased from local shops in the city of Salvador, Bahia. 

### 2.2. Methods 

Below, in Figure 1, is a flowchart of the methods used to develop this work.

#### 2.2.1. Preparation of Aqueous Licuri Extract

The licuri kernels were weighed and immersed in filtered water heated to 100 °C, respecting the 1:3 ratio (kernel:water), for 20 min. The mixture was transferred to a household blender (Mondial Easy Power 550 W) and blended for 5 min at medium speed. The mixture was filtered using a plastic sieve covered with cotton cloth to separate the particulate material. An amount of 10% demerara sugar (*w*/*v*) was then added to the filtrate. 

The aqueous extract of licuri obtained was pasteurized at a temperature of 85 °C for 30 min and cooled to 25 ± 2 °C [23].

#### 2.2.2. Reactivation and Adaptation of Water Kefir Grains in Aqueous Licuri Extract

The water kefir grains, stored at −20 °C, were reactivated in a sucrose solution (10% demerara sugar) containing 2.5% (*w*/*v*) of water kefir grains for 48 h at 25 ± 2 °C in a BOD incubator (QUIMIS, model Q315M16) [25]. A sterile glass container with a capacity of 1 L was used, protected by cotton cloth at the top opening, which was also sterile. Fifteen fermentation cycles were carried out, during which the water kefir grains were separated from the fermented sucrose solution using plastic sieves every 48 h. They were then washed with sterile 0.85% saline solution and inoculated into a new sucrose solution to start a new fermentation cycle (Appendix A) [35].

In the next stage, the grains were adapted to the aqueous licuri extract. In line with the reactivation procedure (48 h/25 °C), the kefir grains were subjected to a further 15 fermentation cycles, using the aqueous licuri extract as a substrate to enable modulation and maintenance of the active grains [36,37,38]. Once this process was complete, the adapted kefir grains were sent to identify the microbial profile and production of the licuri-based fermented kefir beverage. 

#### 2.2.3. Identification of the Microbial Profile of Kefir Grains Adapted in Aqueous Licuri Extract

For DNA extraction, a fraction of the sample was lysed in a disruptor L-BEADER 6 (Loccus, São Paulo, Brazil) using zirconium beads. After cell disruption, the sample was processed using the QiaCube robot (Qiagen, Hilden, Germany). 

For library preparation and sequencing of bacterial 16S DNA, the 341F and 806R oligonucleotides specific to the V3/V4 region of the 16S rRNA gene were used, while for yeasts, amplification of the ITS1 rRNA region was carried out using primers ITS1F5′-GAACCWGCGGARGGATCA-3′ and ITS2R 5′-GCTGCGTTCTTCATCGATGC-3′, as previously described [39,40].

Two microliters of DNA extracted from the sample were used as a template in the first PCR reaction, carried out in Platinum Taq (Invitrogen, Carlsbad, CA, USA) with the following conditions: 95 °C for 5 min, 25 cycles of 95 °C for 45 s, 55 °C for 30 s, 72 °C for 45 s, and a final extension of 72 °C for 2 min for PCR 1. For PCR 2, the conditions were 95 °C for 5 min, 16S 10 cycles, and ITS1 15 cycles of 95 °C for 45 s, 66 °C for 30 s, and 72 °C for 45 s, and a final extension of 72 °C for 2 min. 

The final PCR reactions were purified using Neobeads^®^ (Sera-Mag™ based magnetic beads), and an equivalent volume was added to the sequencing pool. The final DNA concentration of the library pool was estimated by assay using Picogreen dsDNA (Invitrogen, Carlsbad, CA, USA) and then diluted for quantification by qPCR using the Collibri™ Library Quantification Kit (Invitrogen, USA) already optimized for Illumina libraries. The sequencing pool was adjusted to a final concentration of 11 pM (for V2 kits) or 17.5 pM (for V3 kits) and sequenced on the MiSeq system (Illumina, San Diego, CA, USA) using the Illumina sequencing primers supplied with the manufacturer’s kit. Paired-end runs were performed using V2 × 500 or V3 × 600 sequencing kits (Illumina, USA) with 100,000 coverage reads per sample.

Sequences were analyzed using a proprietary pipeline (Neoprospecta Microbiome Technologies, Florianópolis, Brazil). Briefly, all DNA sequences resulting from sequencing were individually passed through a quality filter based on the sum of the error probabilities of their bases, allowing a maximum cumulative error of 1%. The DNA sequences corresponding to the Illumina technology adapters were then removed. Sequences that passed the initial procedures with 100% identity were grouped into phylotypes/clusters and used for taxonomic identification by comparison with a database of accurate ITS sequences (NeoRef (Poitiers, France), Neoprospecta (Florianópolis, Brazil)).

The bioinformatics analysis consisted of reading the quality filter by converting the Q score (QS) to error probability (EP) for each nucleotide using the following Equation (1):(1)EP=10^(−QS)10

Only reads with a sum of errors equal to or less than 1 were considered for downstream analysis. Subsequently, all reads with one or more undetermined “N” bases or cut sequences with two or more consecutive bases with QS less than Q20 were eliminated. Operational Taxonomic Unit (OTU) inference was performed using BLAST 2.2.28 [41] against the Greengenes 13.8 database [42]. For taxonomy assignment, only sequences with success rates of 99% in an alignment covering more than 99% were considered. A value of 2000 bp was used to infer the relevance of the bacterial species identified in the samples obtained.

### 2.3. Preparation of a Fermented Plant-Based Kefir Beverage Made from Licuri Kernel

The aqueous licuri extract was fermented in a BOD incubator at 25 ± 1 °C [20]. Based on preliminary tests, the variable time (24 h and 48 h) and amount of inoculum of water kefir grains adapted to the aqueous extract of licuri (1%, 2.5%, and 5%) were evaluated, as shown in Table 1.

After the fermentation process, the beverages were stored in a refrigerator (DAKO-REDK32) at 4 ± 2 °C, hermetically sealed in glass containers until the time of analysis.

#### Growth of Kefir Grains

To verify the influence of different fermentation times and the amount of inoculum on the percentage growth of kefir grains after the fermentation process of licuri-based beverages, the kefir grains were sieved to separate them from the beverages and washed with sterile saline solution (0.85%). They were then rested on paper towels to partially retain moisture [35].

The kefir grains growth (%) was calculated according to [25] using the following equation:Kefir grains growth (%) = (Fw − Iw)/Iw × 100(2)
where Fw: Final weight (g); Iw: initial weight (g).

### 2.4. Analysis of Quality Parameters in Fermented Plant-Based Kefir Beverages Made from Licuri during Refrigerated Storage

The stability of licuri-based fermented kefir beverages was evaluated using titratable acidity, pH, ethanol content, and the viability of lactic acid bacteria (LAB) and yeasts every seven days (0, 7, 14, 21, and 28 days) under refrigerated storage (4 °C). 

For this purpose, titratable acidity was obtained by titration with 0.1 N sodium hydroxide (NaOH) and expressed in grams of lactic acid/100 mL [43]. The pH was analyzed using a previously calibrated pH meter (Tecnal, model TEC-7, Ourinhos, Brazil). The ethanol content was adapted from [43], where 100 mL of the fermented plant-based licuri beverage was distilled using a rotary evaporator (Quimis, model Q344B2, Diadema, Brazil), and 65 mL of the distillate obtained was taken for density reading on an electronic hydrostatic balance (Gibertini Elettronica™, model Super Alcomat, Novate Milanese, Italy). The ethanol content, expressed as % *v/v*, was obtained from the conversion table of the relative density at 20 °C/20 °C determined in the sample distillate.

The viability of lactic acid bacteria (LAB) was measured on De Man Rogosa and Sharp (MRS) agar (Merck, Darmstadt, Germany) in plates incubated at 37 °C under anaerobic conditions for 72 h. For yeasts, Dicloran Rose-Bengal Chloramphenicol agar (DRCB) culture medium (Merck, Darmstadt, Germany) was used, and the plates were incubated aerobically at 22 °C for 5 days [44]. LAB and yeast viability results were expressed as log CFU/mL of fermented beverage.

#### 2.4.1. Nutritional and Technological Characterization of Fermented Plant-Based Kefir Beverages Made from Licuri Kernel

After fermentation (D0), all the beverages developed were characterized by their proximate composition, color, syneresis, water retention capacity, and rheological behavior.

The proximate composition was carried out according to official methods [40] and expressed in g/100 g. The moisture content was determined by gravimetry in an oven-drying method (Nova Ética, 410ND, São Paulo, Brazil) at 105 °C until constant weight, with subsequent determination of ash by combustion at 550 °C in a muffle oven (Lavoisier, São Paulo, Brazil). Protein content was determined by the Kjeldahl method (chemical digestion, distillation, and titration). Protein content was determined using a conversion factor of 6.25. The lipid content was determined by the Soxhlet method, using petroleum ether (Êxodo Científica, Sumaré, Brazil) as solvent. The carbohydrate content was calculated by difference (100 − moisture − proteins − lipids − ash). To calculate the energy value, the Atwater conversion factor was used, Equation (3) [45].
Energy value = [Protein (g) × 4)] + [Carbohydrate (g) × 4] + [Lipid (g) × 9](3)

A colorimeter (Konica Minolta, model Chroma meter CR-5, Japan) was used for color analysis. The color of the grains and formulations was assessed using the CIELab scale, using the parameters L* (luminosity), a* (red/green), b* (yellow/blue), chroma (C*), and hue (H°). The total color difference (ΔE) between the fermented licuri plant-based kefir beverages, and the aqueous licuri extract was calculated using Equation (4) [46].
ΔE = √ (ΔL*)^2^ + (Δa*)^2^ + (Δb*)^2^(4)
where: L* = (luminosity); a* = (red/green); b* = (yellow/blue).

Syneresis and water retention capacity were carried out using adaptations [22]. For this, an Eppendorf centrifuge, model 5702R (Eppendorf Ltd., Stevenage, UK), was used, where 10 g aliquots of the samples were centrifuged at 3000× *g* for 30 min at 10 °C, and the supernatant was separated. Syneresis (%) was calculated using Equation (5) and water retention capacity using Equation (6).
Syneresis (%) = (Supernatant weight (g))/(Initial weight(g)) ×100(5)
Water retention capacity (%) = (Weight of drained gel (g))/(Initial weight(g)) ×100(6)

Rheological measurements were carried out using a rheometer (Haake Rheotest; Mod. 2.1; Medingen, Germany) with concentric cylinders coupled to a water bath for temperature control (25 °C) and a shear rate between 25 and 1000 s^−1^. Shear stress data measured from the deformation rates were used. The experimental data were fitted to the Ostwald–de Waele model [47] according to Equation (7).
σ = Kγ (n)(7)
where σ is the shear stress (Pa), K is the consistency index (Pa-s^n^), γ is the shear rate (s^−1^), and n is the flow behavior index (dimensionless). 

The rheological data was adjusted to the Ostwald–de Waele model to obtain viscosity values, as in Equation (8).
μ = Kγ (n)^−1^(8)
where μ is the apparent viscosity, K is the consistency index (Pa-s^n^), γ is the shear rate, and N is the flow behavior index. The viscosity results were expressed in mPas.

#### 2.4.2. Microbiological Safety Analysis of Plant-Based Fermented Kefir Beverages Made from Licuri

All samples were analyzed for microbiological safety for molds and yeasts, *Salmonella* spp., total coliforms, thermotolerant coliforms, and *Escherichia coli* according to microorganisms recommended by Codex Stan 243-2003 [48] and methods according to the American Public Health Association [44].

Briefly, 25 mL of each fermented beverage was homogenized in 225 mL of 0.1% peptone water. The beverages were then serially diluted and inoculated into Petri dishes using a Drigalsky loop. Mold and yeast counts were determined according to growth on DRBC agar and incubated aerobically at 25 ± 1 °C for 3–5 days. Colonies were expressed as log colony forming units (CFU) per mL. For the *Salmonella* spp. analysis, the samples were pre-enriched with 1% buffered peptone water (Merck KGaA, Darmstadt, Germany) and incubated at 35 ± 2 °C for 24 h, followed by selective enrichment in Tetrathionate broth (Neogen, Lansing, MI, USA) and Rappaport Vassiliadis broth (RV-Micromed Isofar) (Merck KGaA, Darmstadt, Germany) at 35 ± 0.2 °C for 24 h and 42 ± 2 °C for 24 h, respectively. Subsequently, the aliquots were seeded on Xylose Lysine Deoxycholate (XLD) agar (Neogen, Lansing, MI, USA), Hektoen enteric (HE) agar (Ionlab, Araucária, Brazil), and *Salmonella Shigella* (SS) agar (Neogen, Lansing, MI, USA) and incubated at 35 ± 2 °C for 24 h. Finally, analyses of total (35 °C) and thermotolerant coliforms(45 °C) and *Escherichia coli* were carried out using the most probable number (MPN) technique with Lauryl Sulphate Tryptose (LST) broth (Neogen, Lansing, MI, USA), incubated at 35 ± 0.02 °C for 48 h, and confirmatory analysis of total coliforms in Brilliant Green Bile Broth 2% (CVBB) (Merck) incubated at 35 ± 0.05 °C for 48 h and thermotolerant coliforms in EC Broth (ION) at 35 ± 0.02 °C for 48 h, followed by seeding on Eosin Methylene Blue (EMB) agar for identification of *E. coli*, incubated at 35 ± 1.0 °C for 48 h.

#### 2.4.3. Sensory Analysis—Acceptance Test and Ideal Scale of Fermented Plant-Based Kefir Beverages Made from Licuri Kernel

The research was approved by the Ethics Committee of the Faculty of Pharmacy under CAAE number 69791223.1.0000.8035 (29 August 2023). The sensory analysis was carried out according to the ethical standards for research on human beings (National Health Council, Resolution No. 196/1996 [49]. The judges signed a Free and Informed Consent Form (FICF). A total of 100 untrained tasters (65 women and 35 men) between 18 and 51 years of age were recruited from the Faculty of Pharmacy (Federal University of Bahia, Brazil) and signed the informed consent form.

Furthermore, the beverages’ acidity and sweetness parameters were evaluated by the Just About Right (JAR) test, using an ideal 7-point scale ranging from 1, “much weaker than ideal”, to 7, “much more intense than the ideal” [50]. This test is based on an ideal point model; the JAR level is considered the standard, and if it deviates from the JAR, it implies a variation from the ideal point [51]. 

The tests were carried out in individual booths where the fermented beverages were presented in cups identified with three-digit codes, and 20 mL samples were served monadically at 4 ± 2 °C, with a cup of water for mouth washing between the samples. The results obtained were presented as means.

### 2.5. Data Processing

The analyses were carried out in triplicate (±standard deviation), and the means were evaluated by analysis of variance (ANOVA) and compared using the Tukey test (*p* ≤ 0.05). The correlation between the variables was checked using Spearman’s correlation coefficient, with a significance level of *p* ≤ 0.05. Significant results were then used for principal component analysis (PCA) and hierarchical cluster analysis (dissimilarity: Euclidean distance, Ward agglomeration method, and Hartigan index truncation). 

Statistical calculations were carried out using XLSTAT^®^ software version 2023.1.5 [52]. Origin software version 2017 [53] was used to generate graphs and figures.

## 3. Results and Discussion

### 3.1. Microbial Profile of Water Kefir Grains Adapted in Aqueous Licuri Extract

Metagenomic sequencing was used to identify the bacteria and fungi present in the kefir grains adapted to the aqueous licuri extract, which were then used as inoculum to develop the fermented beverages. The lactic acid bacteria (LAB), acetic acid bacteria (AAB), and yeasts identified in the material analyzed are shown in Table 2, which also shows the number of DNA sequences and the proportions of each species.

The microbial diversity of kefir grains grown in aqueous licuri extract (48 h/25 °C) is mainly composed of lactic acid bacteria (58.27%) and acetic acid bacteria (0.24%). Corroborating this result, Lynch et al. [14] showed that in water kefir grains, it is common for LAB to outnumber yeasts and AAB. 

The LAB identified in the kefir grains were *Lactobacillus hilgardii*, *Lactobacillus casei*, *Lactobacillus ferraginis*, and *Lactobacillus diolivorans*. *Lentilactobacillus* sp. and *Lacticaseibacillus* sp. were only detected at genus level, making it impossible to determine their respective species. Lactic acid bacteria are important for kefir grains because, in general, they are responsible for producing exopolysaccharides by metabolizing glucose and fructose, essential mechanisms for forming the structure of kefir grains [14]. In addition, they are recognized for their probiotic action, and researchers have been engaged in prospecting for these microorganisms [54]. 

*Lactobacillus hilgardii* was the bacteria species identified in the highest percentage, corresponding to 89.24% of the total in this kingdom. This species was also identified in kefir grains grown in sucrose solution [54,55] and in solutions with different proportions of fig extract [35,56,57]; however, its predominance was not as expressive as grains grown in aqueous licuri extract. 

According to the literature [58], *L. hilgardii* is one of the main bacteria responsible for producing exopolysaccharides. Furthermore, according to the findings of Pourramezan, Oloomi, and Kasra-Kermanshah [59], strains of *L. hilgardii* isolated from fermented camel’s milk have demonstrated, through in vitro tests, a potential antioxidant and anticancer effect. 

Thus, due to the strong presence of *L. hilgardii* and other lactic acid bacteria in kefir grains adapted in aqueous licuri extract, a potential probiotic effect of the beverages fermented by them can be speculated. However, since the probiotic effects of microorganisms are strain-dependent [60], studies are needed to evaluate this.

The presence of *Enterococcus hirae* was also detected in the present study, and until then, there were no previous records mentioning the occurrence of this strain in water kefir grains. Although there may be controversies regarding the microbiological safety of the *Enterococcus* sp. genus in food, Carasi et al. [61] demonstrated the safety and functional action of *Enterococcus durans* isolated in milk kefir grains. However, studies are needed to verify the effects of *E. hirae* present in water kefir grains adapted to the aqueous extract of licuri, since there is no information on this to date. 

*Acetobacter orientalis* and *Acetobacter peroxydans* and, at the genus level, *Acetobacter* sp., were the acetic bacteria detected. These microorganisms are part of the kefir microbiota and are responsible for oxidizing organic acids and ethanol [15].

As for the fungi kingdom in water kefir grains adapted to aqueous licuri extract, it is comprised of the genera *Brettanomyces* sp., *Pichiaceae* sp., *Saccharomycetales* sp., *Meyerozyma* sp., and *Starmerella* sp. Although *Saccharomyces cerevisiae* has been reported as the main yeast in water kefir grains [38,39,40,41,42,43,44,45,46,47,48,49,50,51,52,53,54,55,56], the grains adapted to the aqueous licuri extract presented the species *Brettanomyces bruxellensis* as the most predominant, representing a proportion of 59.49% of this community. Bueno et al. [16] also observed a prevalence of this yeast (99.74%) in kefir grains fermented in a sucrose solution (5%). 

All the species identified in this study, except for *E. hirai*, have already been described in other experimental studies with water kefir grains [15,16,17,18,19,20,21,22,23,24,25,26,27,28,29,30,31,32,33,34,35,36,37,38,39,40,41,42,43,44,45,46,47,48,49,50,51,52,53,54,55,56,57,58,59,60,61,62] and are important for the characteristics of the products fermented by them. 

The chemical composition of the substrate is known to influence the selection and prevalence of the microorganisms that comprise a natural fermentation symbiosis [11,12,13,14]. To date, no studies have been described in the literature regarding the evaluation of licuri kernel as a substrate for kefir fermentation or even their prebiotic potential and their derivatives. However, based on these results, it can be inferred that this matrix offers favorable conditions for the growth of potentially probiotic microorganisms, considering the detection of microorganisms of the genus *Lactobacillus* sp., *Lentilactobacillus* sp., and *Lacticaseibacillus* sp. 

### 3.2. Growth of Water Kefir Grains Adapted to Aqueous Licuri Extract

The water kefir grains proved to adapt well to the conditions of the new substrate, showing a firm appearance and a soft, slightly yellowish color (CIELAB: L* 68.19 ± 4.04, a* −0.64 ± 1.73, and b* 2.45 ± 1.86). The percentages of growth obtained, in descending order, for the formulations under different fermentation conditions in aqueous licuri extract were B (894.49 ± 12.92%^a^), D (496.87 ± 4.04%^b^), C (473.14 ± 3.41%^c^), E (304.35 ± 2.24%^d^), A (231.25 ± 5.55%^e^), and F (157.09 ± 1.98%^f^) (Appendix A).

It has been proven that the growth of kefir grains varies according to fermentation time and temperature, substrates, and origin, among others [15,16,17,18,19,20,21,22,23,24,25,26,27,28,29,30,31,32,33,34,35,36,37,38,39,40,41,42,43,44,45,46,47,48,49,50,51,52,53,54,55,56,57,58,59,60,61,62,63]. In the aqueous extract of licuri, it was noted that in conditions where lower percentages of inoculum were fermented for 48 h (B and D), greater (*p* < 0.05) grain growth was obtained. Previously, Laureys et al. [26] also associated higher kefir grain biomass production with lower inoculum concentrations due to reduced acid stress and substrate consumption during the process. 

On the other hand, using a greater quantity of inoculum fermented for 48 h (F) resulted in the lowest production rate. The biomass production of kefir grains in aqueous licuri extract is likely related to the dominance of *Lactobacillus hilgardii* in the inoculum. As mentioned, pioneering studies with water kefir [58] described this species as one of the main producers of exopolysaccharides (α-1,6 branched dextran) in the grains. To do this, an extracellular enzyme, glucansucrase, is secreted, and under optimum pH conditions (4.3–4.6), it synthesizes glucan. However, a pH of around 3.6 is enough to inhibit the activity of this enzyme. 

Considering that after 24 h of fermentation, beverages inoculated with 2.5% grains (E) already had an environment with a pH of 3.76 ± 0.05 (data to be presented), it is likely that the activity of glucansucrase was considerably reduced after another 24 h of fermentation, which may explain the lower percentage of grain growth in formulation F (pH 3.52 ± 0.01). In addition, prolonged exposure of the grains to an acidified medium can damage the structural stability of the grains through acid hydrolysis of the polysaccharides, making them small and brittle [62,63]. 

It was found that the production of biomass using the aqueous extract of licuri as a substrate was considerably higher than the data already reported in the literature, such as in a substrate based on quinoa (64%) [64], honey (40%) [65], coconut (90.89%) [25], sugarcane molasses (19.15%) [66], fig with apricot (4.58%), fig, raisins and lemon (43.98%), fig (68, 82%) [10], and sucrose solution with fig (160%) [26].

Findings in the literature point to acidity as the most important parameter in the growth of water kefir grains. Laureys et al. [63] found that a higher buffering capacity and high calcium concentrations in the fermentation substrate promote a significant increase in grain mass. When these factors are below certain levels, the pH tends to be lower and grain growth decreases, likely due to excessive acid stress. In the same study, it was found that lower concentrations of fermentative inoculum resulted in greater grain growth associated with lower acid stress and substrate inhibition and depletion.

Laureys et al. [57] observed that in kefir samples with greater oxygen availability, acetic acid bacteria predominated. This promoted greater acidification of the environment due to the increased acetic acid content, resulting in reduced kefir grain growth. In addition, the same authors found that high concentrations of nutrients accelerate fermentation, increase metabolites, and maintain stable levels of carbohydrates and pH, which promotes grain growth.

Furthermore, according to Laureys et al. [26], the type of sugar used significantly influences the growth of kefir grains. Partially replacing sucrose with glucose and/or fructose reduces grain growth. Although glucose ferments faster, grain growth is lower with fructose or glucose compared to sucrose. In addition, completely replacing sucrose with glucose and/or fructose results in no growth of kefir grains.

The positive effect of licuri extract on the symbiosis of water kefir grains is undeniable. This finding signals a possible beneficial interaction between these two elements, which may have important implications for the functional aspects of products made from licuri kernel. 

As mentioned, the possible prebiotic properties of licuri kernels and/or their derivatives have not been investigated to date. However, Andrade et al. [67] demonstrated the prebiotic potential of Jerivá kernel, which, like licuri, belongs to the *Syagrus* genus. The authors examined the influence of the pulp and kernel cake of this fruit as a substrate in the fermentation process of three bacterial strains: *Bifidobacterium lactis* BLC1, *Lactobacillus casei* BGP93, and *Lactobacillus acidophilus* LA3, where an increase in bacterial viability was observed, which was associated with the soluble fibers found in the fruit. 

### 3.3. Analysis of Quality Parameters in Fermented Plant-Based Kefir Beverages Made from Licuri during Refrigerated Storage

Table 3 shows results for titratable acidity, pH, and ethanol content during the 28-day storage period (4 ± 2 °C) of the fermented beverages. 

The aqueous extract of licuri had an initial acidity of 0.09 ± 0.00 of lactic acid (mg/100 mL). After the fermentation process, this value varied (*p* < 0.05) between 0.33 ± 0.00 and 0.88 ± 0.00 of lactic acid (mg/100 mL). This is similar to kefir-fermented beverages based on linseed oil cake (0.54 ± 0.01–0.68 ± 0.00) [21], hazelnut (0.33 ± 0.02), peanut (0.40 ± 0.02), walnut (0.25 ± 0.02), coconut (0.49 ± 0.01) [68], and almond (0.47 ± 0.04) [19,20,21,22,23,24,25,26,27,28,29,30,31,32,33,34,35,36,37,38,39,40,41,42,43,44,45,46,47,48,49,50,51,52,53,54,55,56,57,58,59,60,61,62,63,64,65,66,67,68]. 

Beverages fermented for 48 h showed higher (*p* < 0.05) acidity percentages. Similar behavior has been reported in nut extracts [18], corn [69], and apple juice with whey [70], where the longer the fermentation time, the higher the acidity of the resulting beverages. 

Throughout the storage period, all the formulations showed variations (*p* < 0.05) in acidity values. Notably, formulation A, which at T0 had the lowest acidity concentration (*p* < 0.05), showed the greatest increase in the parameter during storage. This can be explained by lower grain growth, i.e., biomass production. According to Laureys et al. [63], lower grain growth implies lower incorporation of glucose into the exopolysaccharide and, consequently, greater availability of substrate for acid production during storage. On the other hand, the formulation has greater stability in refrigerated storage. 

In addition, it was observed that the different fermentation times promoted different behaviors throughout storage. While beverages fermented for 48 h (B, D, and F) showed a reduction (*p* < 0.05) in acidity over the course of the days, the 24 h beverages (A, C, and E) became more acidic. Gocer and Koptagel [19] also observed a reduction in acidity during the storage of nut extracts fermented with kefir and justified this reduction with the metabolization of organic acids by the microorganisms as a form of defense mechanism against the acidic environment. 

It is important to note that although sample A showed a significant increase in acidity over the 21 days, it remained the formulation with the lowest lactic acid content. This result shows that the smaller amount of inoculum and the shorter fermentation time led to a reduced production of lactic acid compared to the other formulations.

To date, there are no specific regulations for plant-based fermented beverages. However, the Codex Alimentarius [48] recommends that the acidity in kefir-fermented milk should be a maximum of 1.00 g of lactic acid/100 g. Therefore, considering this standard, all the beverages remained within the expected range until the 28th day of storage at 4 ± 2 °C. 

The pH of the aqueous licuri extract was 5.93 ± 0.11. After the fermentation, this parameter varied between 3.52 ± 0.01 and 4.29 ± 0.04, similar to water kefir fermentations, which used coconut sugar (3.59 ± 0.01), demerara sugar (3.67 ± 0.03), and cane molasses (3.52 ± 0.03) [66] as substrates. Fiordar et al. [65] state that the reduction in pH after fermentation indicates the good adaptation of the grains to the matrices since the higher acidity results from the metabolization of bacteria and yeasts. 

In D0, the lowest pH values were for formulations B (3.56 ± 0.0), D (3.52 ± 0.01), and F (3.52 ± 0.01), while formulation A presented the highest value for this parameter (4.29 ± 0.04). The pH of all the formulations decreased significantly (*p* < 0.05) during the storage period. Similar results were described for fermented hazelnut extract beverages, where the pH decreased from 4.77 ± 0.06 to 4.43 ± 0.02 at the end of 21 days of storage [22]. 

The fermented licuri beverages produced in 48 h fermentations (B, D, and F) showed better stability for the physicochemical parameters of pH and acidity. The longer fermentation time probably provided maximum use of the available substrates and the production of desirable metabolites [15,16,17,18,19,20,21,22,23,24,25,26]. In addition, it is worth noting that the acidification of the beverages due to these metabolites can also inhibit the growth and proliferation of pathogenic microorganisms, contributing to the safety and preservation of these products over time [16]. 

In this study, all the fermented licuri-based beverages were classified as non-alcoholic during the 28 days of storage, following current regulations [48]. Small concentrations were only found in formulations D, E, and F. These samples showed significantly higher yeast viability than the other formulations, as illustrated in Figure 2b. A negative correlation (Appendix A) between ethanol and carbohydrate has been observed. It may be associated with the ability of yeasts to secrete enzymes capable of hydrolyzing sugars (sucrose, glucose, and fructose) via the glycolytic pathway, producing ethanol [14,15,16,17,18,19,20,21,22,23,24,25,26,27,28,29,30,31,32,33,34,35,36,37,38,39,40,41,42,43,44,45,46,47,48,49,50,51,52,53,54,55,56]. It can, therefore, be inferred that the amount of inoculum and fermentation time for formulations A, B, and C were insufficient to promote ethanol production. 

Low concentrations of ethanol (less than 0.5% *v/v*) were also observed by Bueno et al. [16] and Martínez-Torres et al. [15] in beverages fermented by water kefir. An ethanol content between 0.5% and 1% *v/v* is considered ideal, as this compound adds sensory nuances that contribute to the consumer’s sensory experience [71].

Figure 2 shows the results regarding microbial viability (LAB and yeasts) during the 28 days of storage (4 ± 2 °C) of the fermented licuri beverages. 

The LAB count at D0 was between 8.73 and 9.40 Log CFU/mL, above the minimum quantity of 10^7^ CFU/mL recommended for kefir-fermented milk [48]. In D0, formulations C and F had the highest (*p* < 0.05) concentrations of 9.40 ± 0.08 and 9.24 ± 0.03 Log CFU/mL, respectively. The viability of LAB in all the formulations evaluated is slightly higher than that obtained in other matrices fermented with water kefir, such as soy extract added with inulin [20], fruit juice [46], and linseed cake [21], which described counts between 7 and 8 log CFU/mL. It is also similar to hazelnut milk evaluated by Atalar [22], with counts between 8.61 and 9.07 Log CFU/mL. 

It was possible to observe that throughout storage, there was a decrease (*p* < 0.05) in the viability of LAB in formulations C, D, and F. Chen et al. [71] observed that with increasing fermentation time, viable bacteria and acidity initially increase and then decrease. Initially, the acidity is adequate for the metabolism of the cells and boosts their growth. However, the constant increase in this parameter inhibits them. Variations were observed in all the formulations; however, even after 28 days of storage, all the beverages maintained a viability higher than the recommended standard (10^7^ CFU/mL) [48]. 

Dos Santos et al. [20] observed that the temperature of 7 °C was not enough to mitigate the metabolism of the microorganisms since between times 0 and 14 days, LAB showed growth (<0.05). However, after 14 days, this count was significantly reduced. In fermented licuri beverages, a similar behavior was observed, where at D14, formulations A, B, D, and E showed growth (*p* < 0.05), followed by a reduction at the next points (D21 and D28). However, they remained above the minimum recommended value (10^7^ CFU/mL). 

The means of the yeasts present in fermented licuri beverages were between 4.65 ± 0.02 and 5.34 ± 0.02 Log CFU/mL, higher than the minimum concentration (10^4^ CFU/mL) recommended [48]. 

A significant positive correlation (0.956) (Appendix A) was identified between the amount of inoculum and the concentration (CFU/mL) of yeasts. This result implies a direct association between the initial amount of inoculum and the proportional increase in yeast counts in the evaluated fermented products. 

However, no significance was observed when investigating the correlation between the amount of inoculum and the LAB count. It is plausible to infer that the LAB were less sensitive to variations in the amount of inoculum than yeasts.

According to Gocer and Koptagel [19], plant-based beverages offer a favorable environment for the growth of yeasts and bacteria from kefir grains, even possibly outperforming cow’s milk. Furthermore, the antimicrobial properties of plant matrices such as licuri [31,32,33,34], although playing an important barrier [22,23,24], do not always have significant implications for the survival dynamics of fermentative microorganisms in kefir grains [72]. 

Unlike the LAB behavior, the yeasts tended to grow in all the formulations studied. At the end of the 28 days of storage, they remained above 10^5^ CFU/mL. As previously discussed, yeasts make glucose and fructose available to LAB, which in turn benefits from LAB growth and metabolite production, promoting ideal conditions for yeast growth and development [14]. 

Gocer and Koptagel [19] observed a similar performance, which they attributed to yeasts’ antagonistic action on LAB. Therefore, the synergy between metabolites, LAB, AAB, and yeasts may explain the yeasts’ tendency to grow during storage [14]. 

The substrates used for fermentation are closely related to the stability of the products. For example, it was found that the physicochemical and functional characteristics of a pitaya beverage fermented by water kefir grains remained adequate for only seven days of refrigerated storage (7 °C) [16]. On the other hand, black carrot, blackberry, pomegranate, and strawberry juices fermented by water kefir exhibited stability for 84 days of refrigerated storage (4 ± 0.5 °C) [24]. 

The fermented licuri beverages in this study are the result of spontaneous fermentation processes without the addition of preservative additives and thermal processing after the fermentation process. Therefore, the changes in acidity, pH, and microbial viability presented are to be expected and highlight the importance of evaluating processing and storage conditions to obtain safe, high-quality products throughout storage. 

Considering this, the absence of specific regulatory instructions for fermented plant-based beverages makes it difficult to monitor these products. Therefore, given the increasing demand for plant-based foods, the development of more studies is important to provide a diversity of scientific evidence that, in the short term, will encourage regulatory bodies to draw up norms and guidelines that establish quality, safety, and labeling standards for fermented plant-based beverages.

### 3.4. Nutritional and Technological Characterization of Fermented Plant-Based Kefir Beverages Made from Licuri

Table 4 shows the proximate composition of the aqueous licuri extract and the fermented plant-based licuri beverages. The parameters moisture, lipids, ash, and energy value did not differ (*p* > 0.05) between the formulations studied. 

The mean moisture content was similar to that of a fermented beverage based on bauru kernels, 76.25 ± 0.15 [73], and lower than that found for a coconut beverage with inulin, 82.26 ± 0.32 [25]. 

Lipids were found at concentrations of between 10.36 ± 1.20 and 13.94 ± 2.70 g/100 mL; this value is much higher than that found in coconut-based fermented beverages, which ranged from 2.73 ± 0.04 to 3.26 ± 0.07 [25], in bauru kernel-based beverages of 6.50 ± 1.64 [73], and in hazelnut milk of 1.64 ± 0.02 [22].

This result can be justified by the substantial presence of lipids in licuri kernels, as documented in the literature, which accounts for approximately 49.20 ± 0.08% of the kernels’ composition [28]. 

The carbohydrate percentages for the beverage developed were between 5.86 ± 0.19 and 11.51 ± 1.26 (g/100 g); this difference (*p* < 0.05) between formulations was expected since this is the main substrate used by the microorganisms present in kefir grains during fermentation [63]. These values were similar to those found in coconut plant extracts [20], which ranged from 4.46 ± 0.13 to 13.82 ± 0.43 (g/100 g). 

The fermentation conditions of formulation F provided a slight increase in protein content compared to the aqueous licuri extract and the other formulations. Similar behavior was observed by Łopusiewicz et al. [21] and Alves et al. [25] with significantly higher protein contents after fermentation. Soumya et al. [74] point out that variations in protein content are associated with the activity of microorganisms and variations in the types of fermentable sugars, solids, and the general composition of plant extracts. In this study, the authors observed increased protein content in samples of plant extracts (almond, cashew, peanut, oat, quinoa, and coconut) fermented by *Streptococcus thermophilus* CUD3, corroborating previous observations.

The instrumental color parameters are shown in Table 5. Although the beverages differed (*p* < 0.05) for all the CIELAB parameters, in general, the values indicate a light color (L), close to red (a*) and yellow (b*). Saturation (C*) can be considered moderate, and according to the hue (H°), the dominant color is yellow-green. This profile is similar to that found for extracts made from coconut [25,26,27,28,29,30,31,32,33,34,35,36,37,38,39,40,41,42,43,44,45,46,47,48,49,50,51,52,53,54,55,56,57,58,59,60,61,62,63,64,65,66,67,68,69,70,71,72,73,74,75]. 

The chemical composition contributes to the color characteristics of food products [76]. According to the literature, the main phenolic compounds in licuri kernels are (+)-catechin, (−)-epicatechin, procyanidins B1 and B2, myricetin, quercetin, quercetin-3-O-glucoside, quercetin-3-O-ramnoside, quercetin-3-O-rutinoside, and luteolin [30], and these may be related to the color presented by fermented licuri beverages.

The impact of fermentation on the color profile of foods has already been described in the literature [4]. In the beverages evaluated in this study, it was observed that after fermentation, the formulations showed a significant increase in the luminosity (L*) and saturation (C*) parameters, as well as a greater tendency towards a yellow hue, due to the significant increase in the b* parameter. On the other hand, there was a reduction in the intensity of red coloration (a*). According to the total color difference (∆E*), values found were equal to or greater than 5.18 ± 0.00 [46], which can infer that color changes are noticeable in all formulations. 

It is also worth noting that the samples that were fermented for 48 h (B, D, and F) showed the highest post-fermentation b* (blue-yellow) color indices, indicating a hue closer to yellow. In addition, sample D stood out with the highest (∆E* = 7.69 ± 0.00) among the formulations. As shown in Table 3, in D0, these formulations also stand out for their higher percentage of acidity and lower pH when compared to samples fermented for 24 h. In this way, it can be inferred that the more acidic conditions of these formulations may have promoted greater solubility of the flavonoids present in the raw material studied, resulting in a more pronounced yellow hue [77]. Similar color intensification behavior was also observed in other studies with soy extracts [20], linseed oil [21], and Mediterranean fruits [46]. 

The syneresis parameters, water retention capacity, and rheology parameters of the fermented licuri beverages are shown in Table 6. Figure 3 shows the rheogram of the fermented licuri beverages.

Syneresis refers to the release of water from a gel or semi-solid matrix, forming a separate liquid. In contrast, water retention capacity describes a food’s ability to retain water within its structure [78]. Due to the opposite nature of these parameters, water retention capacity and syneresis are generally negatively correlated.

In this study, the syneresis values were high, ranging from 59.87 ± 1.15% to 70.81 ± 1.24%. Syneresis is characterized as a technological defect that negatively impacts consumer acceptance [78]. However, this limitation is often observed in vegetable beverages, leading to the implementation of stabilizers as a strategy to reduce the associated adverse effects [74]. 

The values observed for water retention capacity were relatively low, ranging from 29.25 ± 0.13% to 39.56 ± 1.59%. According to the literature, the water retention capacity values for fermented soy extract and almonds enriched with *Spirulina* are between 13.35 ± 0.63% and 21.85 ± 1.62% [23], while for coconut milk and almonds, these values are 99.3 ± 0.50% and 91.54 ± 0.50%, respectively [68]. Atalar [22] found that while cow’s milk fermented with kefir has a water retention capacity of 19.49 ± 0.37, mixed samples with 75% hazelnut milk showed 25.75 ± 0.24. According to the same, higher concentrations of hazelnut milk promoted the effective immobilization of the aqueous phase by the hazelnut protein in the kefir network.

Both the syneresis and water retention capacity of a beverage can vary according to the fat content, type of starter culture, production of exopolysaccharides, addition of fibers, extracts, and gums, fermentation temperature, pH of the product, and additives [79]. For the formulations analyzed in this study, except for samples A and B, fermentation significantly reduced syneresis and increased water retention capacity. The sample fermented for 48 h with 5% inoculum had the lowest percentage of syneresis and the highest water retention capacity. 

Some LAB can produce exopolysaccharides, which are used to optimize the syneresis of beverages [80]. In fermented beverages based on licuri, a negative correlation was observed between the amount of inoculum and syneresis. It is, therefore, possible to infer that greater quantities of inoculum may have stimulated the production of exopolysaccharides during the fermentation process, resulting in an improvement in syneresis.

Regarding the beverages’ rheological behavior, the correlation coefficients (R^2^ ≥ 0.990) obtained show that the mathematical model effectively provided accurate adjustments. 

In the formulations evaluated, it was observed (Figure 3a and Figure 2b) that the shear stress increased, and the apparent viscosity of the fluid decreased as the strain rate increased. This behavior indicates that the beverages have non-Newtonian pseudoplastic rheological behavior, which is confirmed by the flow behavior index (n) values shown in Table 6, equal to or less than 0.51 ± 0.04 for all formulations.

This behavior was also found in kefir-fermented extracts based on Brazil nuts, baru, Pará nuts, and macadamia nuts [81], hazelnuts, peanuts, walnuts, and almonds [19].

The fermented beverages’ apparent viscosity and consistency index were significantly higher (*p* < 0.05) than the aqueous licuri extract. While apparent viscosity represents the organization and movement of the components of an emulsion during flow, the consistency index describes the initial resistance to flow [82]. 

According to Gamli and Atasoy [83], during the fermentation of vegetable milk, LAB aggregates the protein filaments into three-dimensional structures through isoelectric precipitation, which gives yogurts a uniform consistency and viscous behavior. In addition, when broken by shear damage, they reduce in size and lower the viscosity of the yogurts formed. Similar behavior has been documented by several researchers [19,20,21,22,23,24,25,26,27,28,29,30,31,32,33,34,35,36,37,38,39,40,41,42,43,44,45,46,47,48,49,50,51,52,53,54,55,56,57,58,59,60,61,62,63,64,65,66,67,68,69,70,71,72,73,74,75,76,77,78,79,80,81,82].

Another relevant factor concerns the exopolysaccharides produced by microorganisms during fermentation. Atalar [22] found that in samples of fermented mixed drinks, higher viscosity and consistency were obtained in samples with a higher percentage of exopolysaccharides. In the drinks evaluated in this study, it is clear that after fermentation, there was an increase in viscosity and consistency, which can also be explained by the presence of this exopolysaccharide.

Thus, it can be said that in licuri-based fermented beverages, it was observed that even without the application of any additives, the products showed attractive nutritional and technological potential for industrial production.

### 3.5. Microbiological Safety Analysis of Fermented Plant-Based Kefir Beverages of Licuri Kernel

In terms of microbiological safety, the results showed no Salmonella or Escherichia coli in any of the formulations. The total coliform count (30 °C) varied between 3 × 100 and 4.16 × 100 MPN/mL, while the thermotolerant coliform count (45 °C) was less than 0.3 MPN/mL in all beverages. The counts for the characteristic mold colonies ranged from 1.1 × 100 to 1.4 × 100 CFU/mL. Detailed results are shown in Appendix A.

There are no specific regulations for vegetable beverages fermented from water kefir grains. However, considering the standards described in Normative Instruction No. 46 [84] for kefir-fermented milk, the microbial counts found in the formulated beverages meet the expected microbiological standards. This indicates that good hygienic and sanitary practices have been followed in the production stage.

### 3.6. Sensory Analysis—Acceptance Test and Ideal Scale of Fermented Plant-Based Kefir Beverages of Licuri Kernel

As shown in Table 7, the different fermentation conditions did not significantly change the acceptability of the beverages evaluated. 

All the formulations had mean scores within the acceptable range (above 5.0). When the parameters appearance, flavor, texture, and global impression were evaluated, all the formulations had mean scores between 6.17 ± 1.94 and 7.16 ± 1.74, which corresponds to an overall good acceptance. 

Regarding the aroma parameter, it showed means corresponding to the neutral point on the hedonic acceptance scale (neither liked nor disliked). The volatile acids produced during fermentation [65] may have impacted the acceptance of the aromas of the beverages developed.

These values are similar to the results found in the literature for fermented beverages based on oilseeds [74], pitaya [16], bauru kernels [73], fruits [46], and nut milk [18]. Thus, the sensory acceptance of fermented licuri beverages aligns with expectations for this type of product. 

Scientific data on the sensory acceptance of products made from licuri is still scarce. However, cereal bars have been made using their kernels [85], and sensory analysis showed that formulations with a higher percentage of licuri had better sensory acceptance. In addition, some of the descriptors used to characterize the bars were sweet aroma, coconut aroma, and intense aroma, as well as flavor descriptors such as residual bitterness, rancid, toasted, and intense coconut. It is known that licuri is genetically similar to coconut (*Cocos nucifera* L.) and, therefore, has a flavor and aroma characteristic of conventional coconut [28].

It can be said that the development of products based on regional foods presents challenges due to their lack of incorporation into conventional products, which limits their consumption. In short, despite being easily accessible, their presence on the market is still restricted. Furthermore, according to Giacalone [86], the sensory acceptance of alternative products to animal-origin foods encompasses several aspects in addition to the inherent sensory characteristics of the food itself. These include the consumer’s familiarity with the product, knowledge of the product’s nutritional and functional characteristics, sociodemographic factors, consumer eating behavior, and genetic markers, among others. 

When the panelists were asked how often they consumed fermented beverages, 57% of the tasters reported consuming them only rarely, 21% reported frequently, 16% reported more than once a month, and 6% reported never having consumed this type of product. Farah, Araújo, and Melo [87] showed that consumers’ lack of familiarity with the product being evaluated can negatively influence the acceptance scores, as these consumers become more critical. Thus, it can be said that the high percentage of infrequent consumers (63%) may have affected the average acceptance of the beverages evaluated since these tasters may not be familiar with the typical taste and texture of fermented beverages.

Table 8 shows the scores for the ideal scale (JAR) and the penalty analysis of the scores for the different formulations of fermented licuri beverages. These results show how the different treatments (time and inoculum) interfered with the judges’ perceptions and acceptance of the acidity and sweetness of the fermented beverages. 

Consumers were able to perceive greater acidity in beverages fermented for 48 h (B, D, and F) (*p* < 0.05) than in beverages fermented for 24 h (A, C, and D). However, it was observed that the different concentrations of inoculum did not lead to significant changes in consumers’ perception of acidity. Corroborating with these results, Cui et al. [18] verified the effect of time and inoculum on the acceptance of nut-based beverages. They observed that more acidic beverages fermented for more than 12 h showed a reduction in acceptance scores due to the more intense taste resulting from the fermentation process. However, the different inoculum concentrations (1, 2.5, and 5%) did not interfere with the sensory perception of the beverages evaluated. In other words, higher inoculum concentrations did not result in more acidic samples. 

It was noted that the perception of sweetness did not vary (*p* > 0.05) for beverages fermented with 1% inoculum (A and B). However, for the other formulations, it was noted that 48 h fermentations were considered insufficiently sweet (*p* < 0.05) by the tasters.

All the formulations were penalized (*p* < 0.05) regarding acidity and sweetness. In general, the results of the penalty analysis corroborate the mean scores obtained on the ideal scale since beverages fermented for 24 h were penalized by the tasters for being less acidic and sweeter than expected. 

However, the beverage fermented for 24 h with 2.5% inoculum (D) behaved differently from the others, with a significant overall penalty, although the two mean decreases (insufficient and too much) were not. The heterogeneity of the judges was noticeable in treatments B and D, where both the insufficient and too much means were significant for sweetness. 

Therefore, although the mean acceptance scores are statistically the same, formulation D has the lowest penalties among the other beverages. In other words, it can be inferred that 48 h of fermentation with 2.5% inoculum results in a beverage with acidity and sweetness that is very close to what consumers expect. 

Analysis of the PCA results (Figure 4) highlighted the quality variables that significantly differentiated the fermented beverages developed in this study. Figure 3 shows the two main components (70.78%), expressed by the x and y axes. The first component explained 42.78% of the variation, while component 2 explained 28.01%.

Cosine square analysis (Appendix A) shows that the variables pH, acidity, ethanol, syneresis, WHC, Pa, carbohydrates, proteins, yeast, ideal scale (acidity and sweetness), and CIELAB color parameters (a*, b*, c*, and L*) are the most important in forming principal components 1 and 2. 

Differentiation patterns between the formulations were observed, and according to the hierarchical agglomerative cluster analysis (Figure 5), they can be grouped into three distinct clusters. Beverages A, C, and E (24 h of fermentation) stand out in component 1, mainly due to their lower acidity and higher JAR sweetness score. 

On the other hand, beverages B and D (fermented for 48 h/1 and 2.5%) are explained by component 2, mainly due to their color (CIELAB) and higher production of kefir grains. Beverage F (48 h/5%) makes up a third group, distinguishing itself from the other formulations mainly due to its higher acidity, rheological behavior, and concentrations of carbohydrates, proteins, and ethanol. 

Thus, the different fermentation conditions resulted in products with different characteristics that did not directly or significantly influence the sensory acceptance scores. However, according to the penalty test, fermentations completed in 24 h have insufficient acidity (JAR) and greater sweetness, and the 48 h fermentation with 5% inoculum results in beverages that are too acidic. Therefore, considering all the other aspects evaluated, it can be suggested that the best condition for producing licuri-based fermented beverages is 48 h of fermentation with 2.5% inoculum (D). 

In addition, licuri kernels are an appropriate matrix for developing fermented plant-based beverages, resulting in beverages with excellent nutritional, microbiological, sensory, chemical, physical, and physicochemical properties. However, further research is still needed to investigate the probiotic potential of this product, including in vitro and in vivo tests, to validate the functional effects of this beverage. 

## 4. Conclusions

We investigated the use of aqueous extract from licuri kernels to produce a beverage fermented by water kefir grains, resulting in high-added-value products. The grains showed an affinity for the aqueous extract of licuri, as evidenced by the mass gain. In these, the microorganisms *Lactobacillus hilfardii* and *Brettanomyces bruxellensis* were predominant. The fermented licuri-based beverages showed similar or superior characteristics to other fermented vegetable beverages reported in the literature. However, it is concluded that the beverages obtained with 48 h of fermentation and a 2.5% (D) inoculum concentration stood out for having a better balance between acidity and sweetness, as shown by the ideal scale test. 

This study has important implications for the food industry. It shows the remarkable performance of licuri kernels as a substrate for kefir grains and their emerging potential to diversify the expanding market for plant-based beverages. These contributions could positively impact the production chain of licuri kernels, generating economic, social, and environmental impacts in the producing regions. 

## Figures and Tables

**Figure 1 foods-13-02056-f001:**
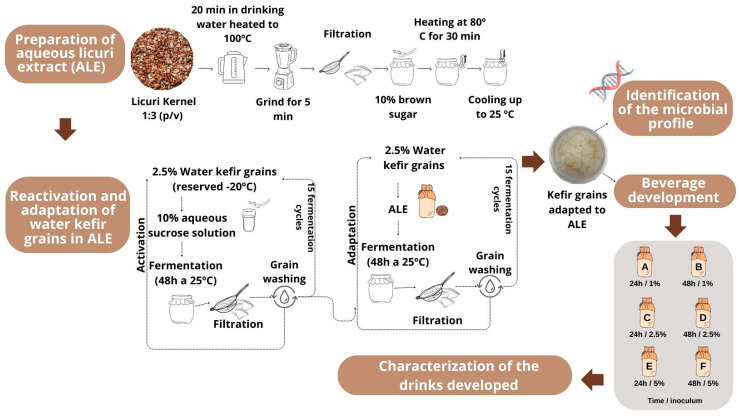
Flowchart outlining the process of aqueous licuri extract preparation, kefir grains adaptation, and fermented plant-based kefir beverages development.

**Figure 2 foods-13-02056-f002:**
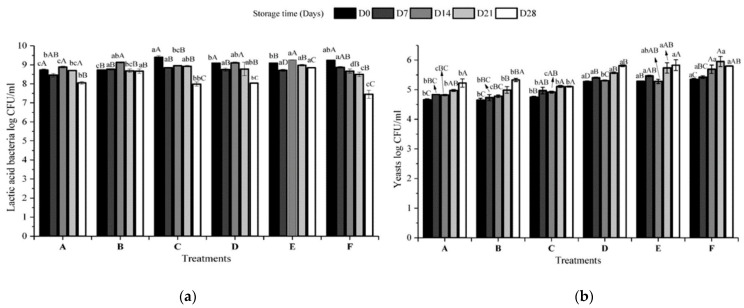
Viability of lactic acid bacteria (**a**) and yeasts (**b**) in plant-based fermented kefir beverages of licuri kernel stored for 28 days at 4 ± 2 °C. Different lowercase letters indicate significant differences (tukey *p* < 0.05) depending on the treatment used for the time evaluated. Different capital letters indicate significant differences (*p* < 0.05) between treatments over storage time. D0: The moment after the grains were removed; D7: After 7 days of storage; DT4: After 14 days of storage; D21: After 21 days of storage; T28: After 28 days of storage. (A) Fermented for 24 h with 1% inoculum; (B) Fermented for 48 h with 1% inoculum; (C) Fermented for 24 h with 2.5% inoculum; (D) Fermented for 48 h with 2.5% inoculum; (E) Fermented for 24 h with 5% inoculum; (F) Fermented for 48 h with 5% inoculum.

**Figure 3 foods-13-02056-f003:**
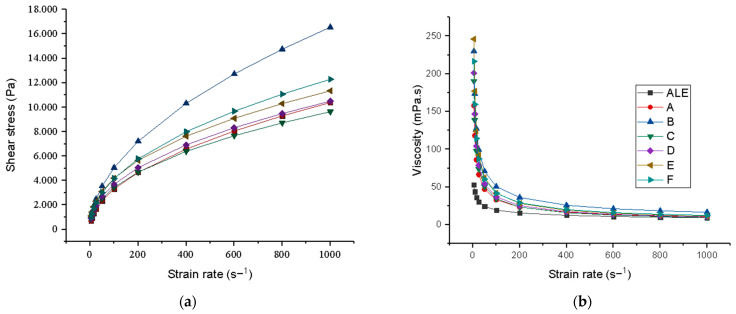
Rheological curve of fermented plant-based kefir beverages of licuri kernel. (**a**) Shear stress versus strain rate; (**b**) Viscosity versus strain rate; (aqueous extract) aqueous licuri extract; (A) fermented for 24 h with a 1% inoculum; (B) fermented for 48 h with a 1% inoculum; (C) fermented for 24 h with a 2.5% inoculum; (D) fermented for 48 h with a 2.5% inoculum; (E) fermented for 24 h with a 5% inoculum; (F) fermented for 48 h with a 5% inoculum.

**Figure 4 foods-13-02056-f004:**
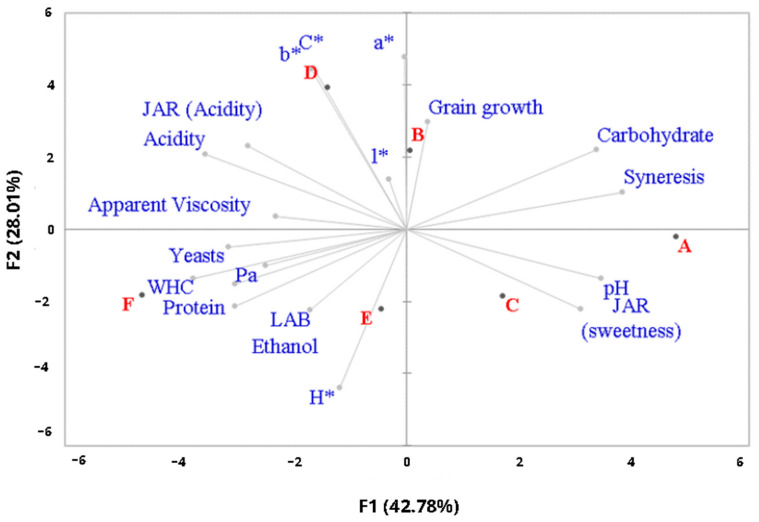
Principal component analysis of parameters of fermented plant-based kefir beverages of licuri kernel. JAR: Mean score obtained on the near-ideal scale for perception of acidity and sweetness; (*) CIELAB system; LAB: lactic acid bacteria; WHC: water holding capacity; (A) fermented for 24 h with 1% inoculum; (B) fermented for 48 h with 1% inoculum; (C) fermented for 24 h with 2.5% inoculum; (D) fermented for 48 h with 2.5% inoculum; (E) fermented for 24 h with 5% inoculum; (F) fermented for 48 h with 5% inoculum.

**Figure 5 foods-13-02056-f005:**
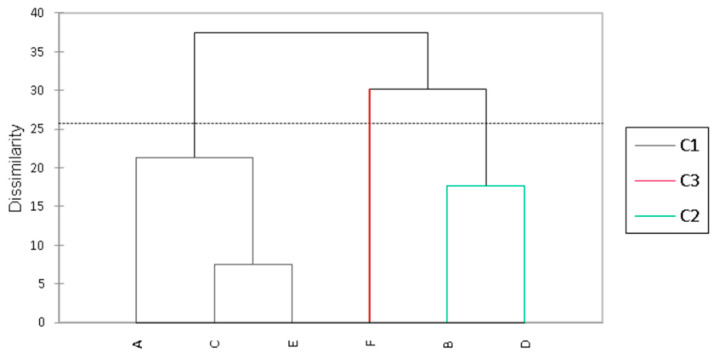
Hierarchical cluster analysis of fermented plant-based kefir beverages of licuri kernel. (A) Fermented for 24 h with 1% inoculum; (B) fermented for 48 h with 1% inoculum; (C) fermented for 24 h with 2.5% inoculum; (D) fermented for 48 h with 2.5% inoculum; (E) fermented for 24 h with 5% inoculum; (F) fermented for 48 h with 5% inoculum. (C1) Cluster 1; (C2) Cluster 2; (C3) Cluster 3.

**Table 1 foods-13-02056-t001:** Formulations of fermented plant-based kefir beverages made from licuri.

Formulations	Variables	
	Fermentation Time (h)	Inoculum (%)
Aqueous licuri extract	0	0
A	24	1
B	48	1
C	24	2.5
D	48	2.5
E	24	5
F	48	5

**Table 2 foods-13-02056-t002:** Microbial diversity of water kefir grains adapted in aqueous licuri extract.

Group	Genus/Species	DNA Sequences	Species (%)	Kingdom (%)
Lactic acid bacteria	*Lactobacillus Hilgardii*	87525	89.24	Bacteria 58.51
	*Lentilactobacillus* sp.	50360	5.13	
	*Lacticaseibacillus* sp.	19930	2.03	
	*Lacticaseibacillus casei*	13160	1.34	
	*Lactobacillus farraginis*	73700	0.75	
	*Lactobacillus diolivorans*	29900	0.30	
	*Enterococcus hirae*	1400	0.01	
Acetic acid bacteria	*Acetobacter* sp.	603	0.61	
	*Acetobacter orientalis*	548	0.56	
	*Acetobacter peroxydans*	8	0.01	
Yeast	*Brettanomyces bruxellensis*	62917	59.49	Fungi 41.49
	*Pichiaceae* sp.	18608	17.59	
	*Brettanomyces anomalus*	11268	10.65	
	*Saccharomyces* sp.	90370	8.54	
	*Brettanomyces* sp.	3776	3.57	
	*Meyerozyma carpophila*	6500	0.06	
	*Pichia kudriavzevii*	57	0.05	
	*Pichia manshurica*	2400	0.02	
	*Starmerela apicola*	600	0.01	

**Table 3 foods-13-02056-t003:** Stability of acidity, pH, and ethanol parameters of fermented plant-based kefir beverages of licuri kernel stored at 4 ± 1 °C.

Parameter	Formulation	D0	D7	D14	D21	D28
Titratable acidity lactic acid (mg/100 mL)	ALE	0.09 ± 0.00 ^g^	****			
	A	0.33 ± 0.00 ^f D^	0.46 ± 0.00 ^eC^	0.46 ± 0.00 ^f BC^	0.47 ± 0.00 ^eB^	0.52 ± 0.00 ^dA^
	B	0.76 ± 0.00 ^cB^	0.80 ± 0.00 ^bA^	0.76 ± 0.00 ^cB^	0.73 ± 0.00 ^bC^	0.71 ± 0.00 ^bD^
	C	0.55 ± 0.00 ^eD^	0.62 ± 0.00 ^dC^	0.65 ± 0.00 ^eB^	0.65 ± 0.00 ^dB^	0.68 ± 0.00 ^cA^
	D	0.88 ± 0.00 ^aA^	0.81 ± 0.00 ^bC^	0.86 ± 0.00 ^aB^	0.80 ± 0.00 ^aC^	0.81 ± 0.00 ^aC^
	E	0.64 ± 0.00 ^dD^	0.69 ± 0.00 ^cC^	0.69 ± 0.00 ^dC^	0.72 ± 0.00 ^cA^	0.70 ± 0.00 ^bB^
	F	0.83 ± 0.00 ^bB^	0.83 ± 0.00 ^aB^	0.84 ± 0.00 ^bA^	0.81 ± 0.00 ^aC^	0.81 ± 0.00 ^aC^
pH	ALE	5.93 ± 0.11 ^a^	**			
	A	4.29 ± 0.04 ^bA^	3.93 ± 0.05 ^aB^	3.87 ± 0.00 ^aBC^	3.80 ± 0.01 ^aC^	3.71 ± 0.01 ^aD^
	B	3.56 ± 0.01 ^dA^	3.55 ± 0.01 ^cA^	3.56 ± 0.00 ^cA^	3.54 ± 0.01 ^cA^	3.48 ± 0.02 ^bB^
	C	3.75 ± 0.01 ^cA^	3.65 ± 0.01 ^bB^	3.64 ± 0.01 ^bB^	3.60 ± 0.00 ^bcC^	3.48 ± 0.01 ^bD^
	D	3.52 ± 0.01 ^dA^	3.50 ± 0.01 ^cA^	3.49 ± 0.01 ^dA^	3.58 ± 0.12 ^bcA^	3.47 ± 0.05 ^bA^
	E	3.76 ± 0.05 ^cA^	3.65 ± 0.01 ^bAB^	3.58 ± 0.01 ^cB^	3.66 ± 0.09 ^abAB^	3.45 ± 0.01 ^bC^
	F	3.52 ± 0.01 ^dAB^	3.53 ± 0.03 ^cA^	3.48 ± 0.02 ^dB^	3.51 ± 0.01 ^cAB^	3.51 ± 0.02 ^bAB^
Ethanol content (% *v*/*v*)	A	*	*	*	*	*
	B	*	*	*	*	*
	C	*	*	*	*	*
	D	*	0.1 ± 0.00 ^aA^	0.1 ± 0.00 ^aA^	0.1 ± 0.00 ^aA^	0.1 ± 0.00 ^aA^
	E	0.1 ± 0.00 ^aA^	0.1 ± 0.00 ^aA^	0.1 ± 0.00 ^aA^	0.1 ± 0.00 ^aA^	0.1 ± 0.00 ^aA^
	F	0.6 ± 0.00 ^aA^	0.6 ± 0.00 ^aA^	0.4 ± 0.00 ^aA^	0.4 ± 0.00 ^aA^	0.3 ± 0.01 ^aA^

(±) Standard deviation of the triplicate analyses used to calculate the means; different lowercase letters indicate significant differences (Tukey *p* < 0.05) depending on the treatment used for the time evaluated. Different capital letters indicate significant differences (*p* < 0.05) between treatments over storage time. D0: The moment after the grains were removed; D7: After 7 days of storage; D14: After 14 days of storage; D21: After 21 days of storage; D28: After 28 days of storage. (A) Fermented for 24 h with 1% inoculum; (B) Fermented for 48 h with 1% inoculum; (C) Fermented for 24 h with 2.5% inoculum; (D) Fermented for 48 h with 2.5% inoculum; (E) Fermented for 24 h with 5% inoculum; (F) Fermented for 48 h with 5% inoculum. * No values detected. ** was not analyzed. ALE stands for aqueous licuri extract.

**Table 4 foods-13-02056-t004:** Proximate composition of aqueous licuri extract and plant-based kefir beverages of licuri kernel, evaluated in the first week of storage (D0).

Formulations	Moisture (g/100)	Lipids (g/100)	Carbohydrates (g/100)	Proteins (g/100)	Ashes (g/100)	Energy Value (Kcal/100 g)
ALE	77.08 ± 0.13 ^a^	10.97 ± 0.18 ^a^	9.98 ± 0.16 ^ab^	1.80 ± 0.37 ^b^	0.23 ± 0.00 ^a^	125.47 ± 29.52 ^a^
A	76.42 ± 0.41 ^a^	10.36 ± 1.20 ^a^	11.51 ± 1.26 ^a^	1.46 ± 0.24 ^c^	0.25 ± 0.00 ^a^	122.11 ± 21.50 ^a^
B	77.10 ± 1.37 ^a^	12.27 ± 0.45 ^a^	9.06 ± 0.50 ^ab^	1.37 ± 0.33 ^c^	0.19 ± 0.01 ^a^	134.04 ± 21.86 ^a^
C	76.98 ± 0.10 ^a^	10.99 ± 1.04 ^a^	10.32 ± 1.07 ^ab^	1.43 ± 0.24 ^c^	0.27 ± 0.00 ^a^	125.31 ± 19.62 ^a^
D	76.22 ± 0.28 ^a^	11.44 ± 0.13 ^a^	10.50 ± 0.08 ^a^	1.55 ± 0.31 ^c^	0.29 ± 0.00 ^a^	130.20 ± 30.62 ^a^
E	76.88 ± 1.07 ^a^	13.72 ± 0.42 ^a^	7.66 ± 0.41 ^bc^	1.49 ± 0.02 ^c^	0.25 ± 0.00 ^a^	144.76 ± 25.46 ^a^
F	75.97 ± 0.30 ^a^	13.94 ± 2.70 ^a^	5.86 ± 0.19 ^c^	2.16 ± 0.84 ^a^	0.27 ± 0.01 ^a^	145.81 ± 7.96 ^a^

(±) Standard deviation of the triplicate analyses used to calculate the means; mean values with different lowercase letters indicate significant differences (*p* < 0.05) between treatments. (A) Fermented for 24 h with a 1% inoculum; (B) Fermented for 48 h with a 1% inoculum; (C) Fermented for 24 h with a 2.5% inoculum; (D) Fermented for 48 h with a 2.5% inoculum; (E) Fermented for 24 h with a 5% inoculum; (F) Fermented for 48 h with a 5% inoculum.

**Table 5 foods-13-02056-t005:** Evaluation of color parameters in fermented plant-based kefir beverages of licuri kernel, evaluated in the first week of storage (D0).

Formulations	L*	a*	b*	C*	H°	∆E*
ALE	75.45 ± 0.02 ^e^	2.59 ± 0.01 ^a^	11.29 ± 0.02 ^g^	11.59 ± 0.02 ^f^	77.05 ± 0.04 ^f^	-
A	81.64 ± 0.04 ^b^	1.60 ± 0.00 ^d^	11.58 ± 0.01 ^e^	11.69 ± 0.01 ^e^	82.14 ± 0.01 ^d^	6.27 ± 0.04 ^c^
B	80.41 ± 0.00 ^d^	1.66 ± 0.00 ^c^	12.50 ± 0.00 ^b^	12.61 ± 0.00 ^b^	82.46 ± 0.00 ^c^	5.18 ± 0.00 ^f^
C	81.63 ± 0.01 ^b^	1.40 ± 0.01 ^f^	11.43 ± 0.02 ^f^	11.51 ± 0.02 ^g^	83.02 ± 0.03 ^b^	6.29 ± 0.01 ^b^
D	82.84 ± 0.01 ^a^	1.96 ± 0.00 ^b^	13.33 ± 0.01 ^a^	13.48 ± 0.01 ^a^	81.64 ± 0.01 ^e^	7.69 ± 0.00 ^a^
E	81.46 ± 0.01 ^c^	1.40 ± 0.01 ^f^	11.67 ± 0.01 ^d^	11.76 ± 0.01 ^d^	83.15 ± 0.03 ^a^	6.13 ± 0.01 ^e^
F	81.48 ± 0.01 ^c^	1.44 ± 0.01 ^e^	12.01 ± 0.01 ^c^	12.10 ± 0.01 ^c^	83.13 ± 0.02 ^a^	6.18 ± 0.01 ^d^

(±) Standard deviation of the triplicate analyses used to calculate the means; mean values with different lowercase letters indicate significant differences (*p* < 0.05) between treatments. (L*) luminosity; (a*) red/green; (b*) yellow/blue; (c*) chroma; (h°) hue; (ΔE) total color difference; (A) Fermented for 24 h with a 1% inoculum; (B) Fermented for 48 h with a 1% inoculum; (C) Fermented for 24 h with a 2.5% inoculum; (D) Fermented for 48 h with a 2.5% inoculum; (E) Fermented for 24 h with a 5% inoculum; (F) Fermented for 48 h with a 5% inoculum. ALE stands for aqueous licuri extract.

**Table 6 foods-13-02056-t006:** Syneresis, water retention capacity, and rheology parameters in fermented plant-based kefir beverages of licuri kernel, evaluated in the first week of storage (D0).

Formulation	Syneresis (%)	WHC (%)				
			N	(mPa∙^s^)	K (Pa.s ^n^)	R^2^
ALE	70.43 ± 1.61 ^a^	30.13 ± 1.55 ^cd^	0.67 ± 0.01 ^a^	30.39 ± 1.57 ^e^	85.79 ± 6.07 ^c^	0.99
A	70.81 ± 1.24 ^a^	29.25 ± 0.13 ^d^	0.49 ± 0.02 ^bc^	66.50 ± 5.24 ^d^	337.72 ± 47.83 ^b^	0.99
B	66.86 ± 0.43 ^ab^	32.17 ± 0.11 ^bcd^	0.51 ± 0.04 ^b^	99.41 ± 8.52 ^a^	481.65 ± 95.02 ^ab^	0.99
C	65.20 ± 1.13 ^b^	33.91 ± 0.87 ^bc^	0.45 ± 0.01 ^cd^	73.82 ± 2.88 ^cd^	436.55 ± 26.86 ^ab^	0.98
D	63.64 ± 0.51 ^bc^	34.63 ± 0.67 ^b^	0.45 ± 0.01 ^cd^	78.73± 1.67 ^bcd^	457.37 ± 22.06 ^ab^	0.97
E	63.62 ± 1.13 ^bc^	34.54 ± 1.27 ^b^	0.43 ± 0.03 ^d^	92.41 ± 11.46 ^ab^	582.82 ± 130.50 ^a^	0.98
F	59.87 ± 1.15 ^c^	39.56 ± 1.59 ^a^	0.47 ± 0.00 ^bcd^	87.13± 0.82 ^abc^	481.21 ± 3.59 ^ab^	0.97

(±) Standard deviation of the triplicate analyses used to calculate the means; mean values with different lowercase letters indicate significant differences (*p* < 0.05) between treatments. (A) Fermented for 24 h with 1% inoculum; (B) fermented for 48 h with 1% inoculum; (C) fermented for 24 h with 2.5% inoculum; (D) fermented for 48 h with 2.5% inoculum; (E) fermented for 24 h with 5% inoculum; (F) fermented for 48 h with 5% inoculum. n: flow behavior index; K (Pa.s ^n^): consistency index; (mPa∙^s^) apparent viscosity (measured at 25 °C at a shear rate of 25 s^−1^; R^2^: linear correlation coefficient. ALE stands for aqueous licuri extract.

**Table 7 foods-13-02056-t007:** Score of the attributes evaluated in the sensory acceptance test of fermented plant-based kefir beverages of licuri kernel.

Formulations	Appearance	Aroma	Flavor	Texture	Global Impression
A	6.34 ± 1.96 ^a^	5.89 ± 1.96 ^a^	6.84 ± 1.89 ^a^	7.16 ± 1.74 ^a^	6.95 ± 1.64 ^a^
B	6.31 ± 1.87 ^a^	5.35 ± 2.07 ^a^	6.17 ± 2.20 ^a^	6.71 ± 1.89 ^a^	6.31 ± 1.90 ^a^
C	6.42 ± 1.86 ^a^	5.61 ± 1.95 ^a^	6.54 ± 1.96 ^a^	6.51 ± 1.99 ^a^	6.51 ± 1.67 ^a^
D	6.44 ± 1.67 ^a^	5.59 ± 2.02 ^a^	6.20 ± 2.28 ^a^	6.7 ± 1.97 ^a^	6.36 ± 1.98 ^a^
E	6.17 ± 1.94 ^a^	5.55 ± 2.01 ^a^	6.62 ± 1.94 ^a^	6.74 ± 1.97 ^a^	6.66 ± 1.55 ^a^
F	6.49 ± 2.00 ^a^	5.53 ± 1.94 ^a^	6.19 ± 2.22 ^a^	6.82 ± 1.73 ^a^	6.44 ± 1.87 ^a^

Mean values with different lowercase letters indicate significant differences (*p* < 0.05) between treatments. (A) Fermented for 24 h with 1% inoculum; (B) Fermented for 48 h with 1% inoculum; (C) Fermented for 24 h with 2.5% inoculum; (D) Fermented for 48 h with 2.5% inoculum; (E) Fermented for 24 h with 5% inoculum; (F) Fermented for 48 h with 5% inoculum.

**Table 8 foods-13-02056-t008:** Scores for the ideal scale (JAR) and score penalty analysis of fermented plant-based kefir beverages of licuri kernel.

Formulations	Ideal Scale		Penalty Analysis (% of Consumers and Mean Drop)			
	Acidity	Sweetness	Acidity		Sweetness	
			Insufficient	Too Much	Insufficient	Too Much
A	3.52 ± 1.00 ^b^	4.52 ± 1.4 ^ab^	35.00 (3.84)	-	-	44.00 (2.31)
B	4.62 ± 1.70 ^a^	4.09 ± 1.21 ^bc^	-	50.00 (3.34)	22.00 (2.94)	30.00 (2.40)
C	3.51 ± 1.23 ^b^	4.64 ± 1.27 ^a^	43.00% (3.64)	-	-	47.00 (3.90)
D	4.36± 1.30 ^a^	4.02± 1.08 ^c^	20.00 (2.04) *	41.00 (1.52) *	22.00 (7.05) *	23.00 (4.59)
E	3.34 ± 1.22 ^b^	4.58 ± 1.20 ^a^	48 (2.88)	-	-	47 (3.68)
F	4.80 ± 1.3 ^a^	3.81 ± 1.02 ^c^	-	58 (3.41)	30.00 (4.52)	-

Mean values with different lowercase letters indicate significant differences (*p* < 0.05) between treatments; * not significant (*p* > 0.05); (-) indicates that less than 20% of consumers chose this JAR category; the number in brackets indicates the mean drop, calculated by subtracting the group’s overall mean JAR acceptance from the group’s mean global acceptance. (A) Fermented for 24 h with 1% inoculum; (B) fermented for 48 h with 1% inoculum; (C) fermented for 24 h with 2.5% inoculum; (D) fermented for 48 h with 2.5% inoculum; (E) fermented for 24 h with 5% inoculum; (F) fermented for 48 h with 5% inoculum.

## Data Availability

The original contributions presented in the study are included in the article; further inquiries can be directed to the corresponding author.

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
