# Peer review of "Licuri Kernel (Syagrus coronata (Martius) Beccari): A Promising Matrix for the Development of Fermented Plant-Based Kefir Beverages"

_foods, 2024, doi:10.3390/foods13132056_

Round 1
Reviewer 1 Report
Comments and Suggestions for Authors
The submitted study focuses on the production of plant-based kefir using the licuri kernels as the raw material. For this purpose, the authors first investigated the adaptation and potential use of water kefir in the aqueous licuri extract. As they ensured microbial fermentation, they developed 6 types of kefirs with variable incubation times and inoculum rates. These formulations were tested for many quality parameters. One of them was viability of lactic acid bacteria and yeasts in plant-based fermented kefir beverages of licuri kernel stored for 28 days at 4 ±2 °C, which has proved the viability of kefir cultures during its storage.
The title of this study fits well with the concept of this manuscript. However, abstract needs to be developed to give more info about the quality of product.
INTRODUCTION
The motivation of the study is not clear. There are two questions related to the introduction section. 1. What was the reason for choosing licuri as the raw materials of plant-based kefir among many other potentials? 2. What was the motivation of feeding the licuri to kefir fermentation among many other plant-based processing methods?
What is the production amount of licuri per year? Or any other indicator/information that gives idea about the sustainable supply of licuri is also welcomed.
Lines 72-73: Objective sentence should be revised with more details in line with the content of the study.
MATERIALS & METHODS
This section includes sufficient information on the details of performed analysis/production methods.
RESULTS & DISCUSSIONS
The discussions in microbial profile of water kefir grains adapted in aqueous licuri extract and growth of water kefir grains adapted to aqueous licuri extract sections are satisfactory.
L 446-449: the possibility of post-acidification was well discussed.
L569: why do you think the formulation F improved the protein content of the product? In the text the reason was justified as the growth of microorganisms, however Fig. 2 shows that number of bacteria is not higher than the others.
Please indicate the meanings of F1 to F5 in Table S1. Cosine of Squares (ACP).
CONCLUSION
The conclusions were made depending on the obtained results. This section also includes the possible benefits of the production.
Please kindly find my extra comments in the comments part of the uploaded pdf.

Author Response
File in attachment

Reviewer 2 Report
Comments and Suggestions for Authors
Manuscript 3040197
Journal Foods
Title Licuri kernel (Syagrus coronata (Martius) Beccari): a promising matrix for the development of fermented plant-based kefir beverages
The manuscript entitled “Licuri kernel (Syagrus coronata (Martius) Beccari): a promising matrix for the development of fermented plant-based kefir beverages” describes the development of a plant-based beverage based on the use of kefir grains and licuri extract. Metagenomic analysis was carried out on kefir grains, while the quality parameters, nutritional and technological characteristics of the beverage were evaluated during cold storage. The manuscript is interesting but several parts need revision. Please follow the comments in the file.

Moderate changes are necessary
Author Response
File in attachment

Round 2
Reviewer 1 Report
Comments and Suggestions for Authors
I have gone through each response of authors. I have also checked the responses embedded in the manuscript. As the Reviewer, I want to inform you that the Authors have successfully responded or revised the manuscript according to all my previous comments.
I think that as a result of this review process, the manuscript was developed enough to be interesting to the Readers of Foods.
Author Response
Dear,
We would like to express our sincere appreciation for your valuable contribution to reviewing our manuscript. Your suggestions and corrections were fundamental to enhancing the quality of our work.
We are very pleased to know that the manuscript now meets the requirements of the renowned journal. We thank you immensely for your time and dedication.
Sincerely,
The Authors
Reviewer 2 Report
Comments and Suggestions for Authors
Authors revised the original manuscript but several comments were not correctly addressed. A major revision is still reccomended. Please follow the comments below:
L46-47 Please describe the antinutritional compounds of these products
L52-53 Please expand this part on the effect of fermentation on sensory properties of plant-based beverages. The papers doi.org/10.3390/foods11213346 and doi.org/10.1016/j.cofs.2022.100919 are suggested for your discussion.
L54 Which is the source of water kefir grains? How they are produced? Please add this information
L161-165 Please add the bioinformatic pipeline and the database. Rewrite this part. Additional information on the processing of reads are necessary. See metagenomic studies in literature
L198 Please add a formula to obtain the ethanol content
L212-214 Please better describe these methods. Revise
L262-263 Add °C
L393-395 Which is the specific effect of these parameters? Please be specific. Revise
L645-646 Rewrite. It is not correct in English
Figure 3,4,5 Move the captions under the figures. Revise
L717-718 Describe the results of Table S2 in this section
L893-1122 Revise the reference list. Some references are not correct. Please see the guidelines of the Journal.
Comments on the Quality of English Language
Moderate changes are necessary
